# PPT: Adaptive Token Pruning and Pooling for Efficient Vision Transformers

## Abstract

Vision Transformers (ViTs) have emerged as powerful models in the field of computer vision, providing superior performance in various vision tasks. However, the high computational complexity poses a significant barrier to practical applications in real-world scenarios. Motivated by the fact that not all tokens contribute equally to the final predictions and fewer tokens bring less computational cost, reducing redundant tokens has become a prevailing paradigm for accelerating vision transformers. However, we argue that it is not optimal to either only reduce inattentive redundancy by token pruning or only reduce duplicative redundancy by token merging. To this end, in this paper, we propose a novel acceleration framework, namely token Pruning & Pooling Transformers (PPT), to adaptively tackle these two types of redundancy in different layers. By heuristically integrating both token pruning and token pooling techniques in ViTs without additional trainable parameters, PPT effectively reduces the complexity of the model while maintaining its predictive accuracy. For example, PPT reduces over 37% FLOPs and improves throughput by over 45% for DeiT-S without any accuracy drop on the ImageNet dataset.

## 1 Introduction

In recent years, vision transformers (ViTs) have demonstrated promising results in many vision tasks such as image classification (Dosovitskiy et al., 2021; Jiang et al., 2021; Touvron et al., 2021; Yuan et al., 2021), object detection (Carion et al., 2020; Dai et al., 2021; Li et al., 2022a; Zhu et al., 2021) and semantic segmentation (Kirillov et al., 2023; Liu et al., 2021; Wang et al., 2021). Compared with the convolutional neural networks (CNNs), ViTs have the property of modeling long-range dependencies with the attention mechanism (Vaswani et al., 2017), which introduces fewer inductive biases hence has the potential to absorb more training data. However, densely modeling long-range dependencies among image tokens can lead to computational inefficiency, especially when dealing with large datasets and training iterations (Carion et al., 2020; Dosovitskiy et al., 2021). This in turn limits the further implementation of ViTs in real-world scenarios.

Given the strong correlation between model complexity and the number of tokens in ViTs, a direct approach to accelerate ViTs is to reduce the number of redundant tokens. Moreover, some studies also showed that not all tokens contribute equally to the final predictions (Caron et al., 2021; Pan et al., 2021). Existing attempts to achieve token compression mainly consist of two branches of solutions, *i.e.*, token pruning and token pooling. The former approach emphasizes the design of different importance evaluation strategies to identify and retain relevant tokens while discarding irrelevant ones, as demonstrated in previous research (Fayyaz et al., 2022; Liang et al., 2022; Rao et al., 2021; Xu et al., 2022; Tang et al., 2022; 2023). The latter technique focuses mainly on merging similar image tokens using a predefined similarity evaluation metric and merging policy (Bolya et al., 2023; Marin et al., 2021). To summarize, there are two types of redundancy in vision transformers, *i.e.*, **inattentive redundancy** and **duplicate redundancy**. However, it should be noted that the aforementioned each method only addresses one type of redundancy, *i.e.*, token pruning for inattentive redundancy while token pooling for duplicative redundancy. We argue that reducing only one type of redundancy leads to suboptimal acceleration performance.

In this paper, we propose a novel framework, named as token Pruning & Pooling Transformers (PPT), to jointly tackle the two types of redundancy as shown in Figure 1. Our research investigates that the importance of image tokens becomes more distinct as the layer deepens, indicating that applying token pruning techniques at deeper layers is more suitable. In contrast, the model prefers to use token pooling methods in shallow layers, where a large number of tokens exhibit relatively high similarity. Furthermore, we observe that the distribution of token scores varies between different samples within the same layer. Therefore, we propose an instance-aware adaptive strategy to automatically choose the optimal policy of token pruning or pooling in different layers. Moreover, our method introduces *no trainable parameters*. This indicates that it can be easily integrated into pre-trained ViTs with minimal accuracy degradation, and fine-tuning with PPT can

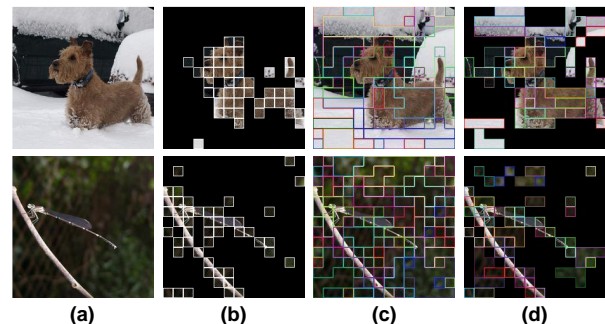

Figure 1: Visualizations of token compression results using different methods on the ImageNet dataset with DeiT-S. (a) Original images. (b) Token pruning methods, which discard inattentive tokens. (c) Token pooling methods, which merge similar tokens within the same color bounding box. (d) Our method can effectively address both types of redundancy while achieving superior performance.

lead to improved accuracy and faster training speed, making it especially beneficial for huge models. Our method is extensively evaluated for different benchmark vision transformers on ImageNet dataset. Experiment results demonstrate that our method outperforms state-of-the-art token compression methods and achieves superior trade-off between accuracy and computational.

We summarize the main contributions as follows: (1) We propose PPT, which is a heuristic framework that acknowledges the complementary potential of token pruning and token merging techniques for effiffient ViTs. (2) We design a redundancy criterion to guide adaptive decision-making on prioritizing different token compression policies for various layers and instances. (3) Our method is simple yet effective and can be easily incorporated into the standard transformer block *without additional trainable parameters*. (4) We perform extensive experiments and obtatin promising results for several different ViTs, *e.g.*, PPT can **reduce over 37% FLOPs** and **improve the throughput by over 45%** for DeiT-S **without any accuracy drop** on ImageNet. We hope our PPT could bring a new perspective for obtaining efficient vision transformers.

## 2    Related Work

**Vision Transformers.** The success of Transformers in natural language processing (Brown et al., 2020; Kenton & Toutanova, 2019; Vaswani et al., 2017) has inspired their application to computer vision, leading to the development of Vision Transformers (ViTs) (Dosovitskiy et al., 2021), which have shown promising results. However, ViTs require large-scale datasets such as ImageNet-22K and JFT-300M for pretraining, as well as substantial computational resources. To address these challenges, DeiT (Dosovitskiy et al., 2021) optimized training strategies and introduced a distilled token to enhance knowledge transfer from a teacher network, significantly improving training efficiency. Following works have focused on incorporating local dependencies into ViTs by modifying patch embeddings or transformer blocks, thereby improving performance:

- *Patch Embedding and Local Dependencies*: Token-to-Token (T2T) ViT (Yuan et al., 2021) progressively merges neighboring tokens into a single token, which reduces token count and enhances spatial context aggregation. Transformer in Transformer (TNT) (Han et al., 2021) introduces a dual-level attention mechanism that includes both patch-level and local sub-patch-level attention, which further enriches local feature representation. The Token Labeling ViT (LV-ViT) (Jiang et al., 2021) utilizes local information embedded in patch tokens with a novel token labeling training objective.

- *Hierarchical Multi-Stage Designs*: Multi-stage hierarchical designs for ViTs have also proven effective, where the number of tokens is progressively reduced while token feature dimensions increase, producing strong features for dense prediction tasks. Pyramid ViT (PVT) (Wang et al., 2021) was the first hierarchical ViT, proposing a progressive shrinking pyramid and spatial-reduction attention mechanism. Swin Transformer (Liu

et al., 2021) adopts a multi-stage hierarchical architecture, computing attention within a local window and using a shifted window partitioning strategy to capture interactions between different image regions.

Despite the advancements in ViTs and their variants, the high computational costs remain a challenge for practical implementation. The main source of computational cost is the quadratic complexity of the multi-head self-attention mechanism with respect to the number of tokens, as well as the linear complexity of the feed-forward network. Therefore, token compression has emerged as a popular approach to accelerate ViTs by reducing the computational burden while retaining expressive power and model performance.

**Token Pruning.** As a main branch of token compression, token pruning aims to retain attentive tokens and prune inattentive ones by designing importance evaluation strategies. In Tang et al. (2022), the authors adopt a top-down paradigm to estimate the impact of each token and remove redundant ones. The IA-RED$^2$ (Pan et al., 2021) is designed to reduce input-uncorrelated tokens hierarchically, while also taking into account interpretability. DynamicViT (Rao et al., 2021) introduces lightweight prediction modules to score tokens and discard unimportant tokens. Evo-ViT (Xu et al., 2022) and EViT (Liang et al., 2022) show that the attention scores between classification tokens and image tokens can be utilized for importance assignment. STViT Chang et al. (2023) leverages the clustering property of self-attention to generate a few semantic tokens representing high-level information, replacing redundant image tokens. A-ViT (Yin et al., 2022), ATS (Fayyaz et al., 2022) and DiffRate Chen et al. (2023) go further by dynamically adjusting the *pruning rate* based on the complexity of the input image. However, mainstream deep learning frameworks do not fully support dynamic token-length inference during batch processing. While token pruning methods achieve promising performance, we realize that they pay less attention to redundancy in the foreground region.

**Token Pooling.** On the other hand, there are some attempts to recognize the importance of pooling tokens together. To alleviate information loss, Evo-ViT (Xu et al., 2022), EViT (Liang et al., 2022), and TPS Wei et al. (2023) merge the tokens they pruned into a single token. To improve the efficiency of ViTs, *Token Pooling* (Marin et al., 2021) utilizes a K-Means-based clustering approach to exploit redundancies in the images, while ToMe (Bolya et al., 2023) thoroughly investigates token similarity and proposes a Bipartite Soft Matching algorithm (BSM) to gradually merge similar tokens. BAT Long et al. (2023) emphasizes the importance of diverse global semantics and propose an efficient token decoupling and merging method that considers both token importance and diversity for pruning. MCTF Lee et al. (2024) gradually fuses the tokens based on multi-criteria and utilizes the one-step-ahead attention to acchieve better performance. In Yuan et al. (2024), the authors address optimization challenges in token merging with a set of techniques, including soft token merging, information-preserving inflation, and parameter-efficient tuning. Token pooling methods can reduce the duplicative redundancy to a certain extent, but they do not take into account that not all tokens contribute equally to the final prediction.

## 3 Preliminaries

In this section, we briefly review the formulation of standard ViTs Dosovitskiy et al. (2021). In ViTs, the input image is split into $N$ patches, and each patch is linearly projected into a latent embedding. An additional classification token is introduced to the set of image tokens, which aggregates global image information and is responsible for the final classification. All tokens are augmented by learnable positional encodings and fed into stacked Transformer encoder blocks, consisting of a multi-head self-attention (MHSA) layer and a feed-forward network (FFN).

In the MHSA layer Dosovitskiy et al. (2021), the input tokens $\mathcal{I} \in \mathbb{R}^{(N+1)\times d}$ are first projected into matrices $\mathcal{Q}$, $\mathcal{K}$, and $\mathcal{V}$, corresponding to queries, keys, and values, respectively. The first row of these matrices represents the classification token, while the remaining $N$ rows correspond to image tokens.

For multi-head self-attention, the input is projected into $H$ different heads, each with its own set of queries, keys, and values:

$$\mathcal{Q}_h = \mathcal{I}\mathbf{W}_{Q_h}, \quad \mathcal{K}_h = \mathcal{I}\mathbf{W}_{K_h}, \quad \mathcal{V}_h = \mathcal{I}\mathbf{W}_{V_h}, \quad h \in \{1,\ldots,H\}, \tag{1}$$

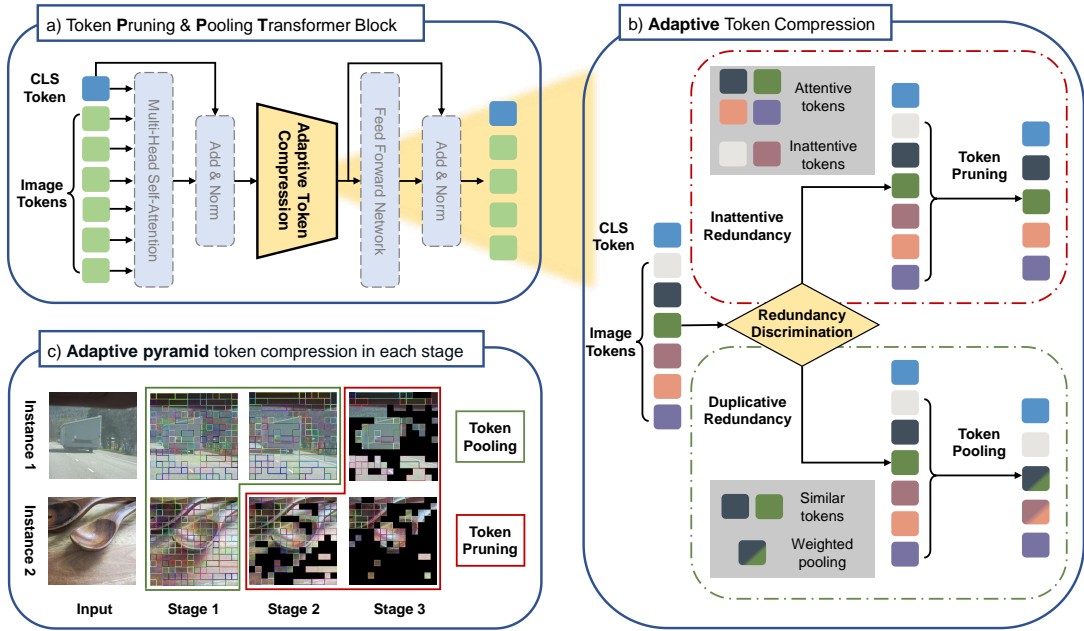

Figure 2: **Overview of the proposed PPT approach**. (a) The module **Adaptive Token Compression** is simple and can be easily inserted inside the standard transformer block *without an additional training parameter*. (b) Our module can **adaptively** select either token pruning **or** token pooling policy to tackle corresponding redundancy based on the current token distribution, which is intuitively reflected across various instances and layers in (c). (c) With PPT, similar patches within the same color bounding box are pooled into a single token, while the masked inattentive patches are pruned, resulting in promising trade-offs between the accuracy and FLOPs.

where $\mathbf{W}_{Q_h}, \mathbf{W}_{K_h}, \mathbf{W}_{V_h} \in \mathbb{R}^{d \times d_h}$ are learnable projection matrices, and $d_h = d/H$ is the dimensionality of each head. The attention matrix for each head and the output for each head are calculated as:

$$\mathcal{A}_h = \text{Softmax}\left(\frac{\mathcal{Q}_h \mathcal{K}_h^T}{\sqrt{d_h}}\right), \quad \text{head}_h = \mathcal{A}_h \mathcal{V}_h. \tag{2}$$

The outputs from all heads are then concatenated and linearly transformed to produce the final output of the MHSA layer:

$$\mathcal{O} = \text{Concat}(\text{head}_1, \text{head}_2, \ldots, \text{head}_H)\mathbf{W}_O, \tag{3}$$

where $\mathbf{W}_O \in \mathbb{R}^{d \times d}$ is the output projection matrix. The output tokens $\mathcal{O} \in \mathbb{R}^{(N+1) \times d}$ are then passed through the FFN layer, which consists of two fully connected linear layers with an activation function. Residual connections He et al. (2016) and layer normalization are applied around both the MHSA and FFN layers. At the final Transformer encoder layer, the classification token is extracted for object category prediction. More details of ViT and the attention mechanism can be found in Dosovitskiy et al. (2021); Vaswani et al. (2017).

# 4 Methodology

## 4.1 Overview

Compared to existing works, our method comprehensively takes into account inattentive redundancy and duplicative redundancy in images. To address these issues, we heuristically integrate both token pruning and token pooling techniques, resulting in favorable trade-offs between the accuracy and FLOPs of ViTs.

As shown in Figure 2, our approach can adaptively select either the token pruning or the token pooling policy based on the current token distribution, which is reflected across different inputs and layers. Furthermore, our method does not involve any trainable parameters, which makes it suitable *with or without training*. It can achieve impressive results even in *off-the-shelf* scenarios, where no customization or fine-tuning is

required. In this section, we first introduce the token pruning and token pooling techniques utilized in our method (Section 4.2 and Section 4.3), and then describe how we integrate them to achieve adaptive token compression in detail (Section 4.4).

## 4.2 Token Pruning for Inattentive Redundancy

In general, the token pruning paradigm consists of two steps, token scoring and token selecting. Token scoring assigns a score to each token based on its importance to the task, and then token selection determines which tokens to keep and which to discard.

**Token Scoring.** Due to only the classification token directly affecting the final prediction, we focus mainly on it. The calculation of the output classification token $\mathcal{O}_{cls}$ can be directly derived as follows:

$$\mathcal{O}_{cls} = \sum_{h=1}^{H} \sum_{i=1}^{N+1} \mathcal{A}_{1,i}^{(h)} \times \mathcal{V}_i^{(h)}, \tag{4}$$

where $H$ is the number of attention heads, and $\mathcal{A}_{1,i}^{(h)}$ and $\mathcal{V}_i^{(h)}$ represent the attention weights and value vectors for head $h$. As we can see, $\mathcal{O}_{cls}$ involves the weighted sum of values of all $N+1$ tokens. The weights are determined by the attention between the classification token and the other tokens. Intuitively, the norm of the weighted value reflects the contribution and importance of the token. Therefore, the significance score of image token $i$ is given by averaging across all heads:

$$\text{Score}_i = \frac{1}{H} \sum_{h=1}^{H} \frac{\mathcal{A}_{1,i+1}^{(h)} \times \left\| \mathcal{V}_{i+1}^{(h)} \right\|}{\sum_{j=1}^{N} \mathcal{A}_{1,j+1}^{(h)} \times \left\| \mathcal{V}_{j+1}^{(h)} \right\|}, \quad i,j \in \{1, \dots, N\}. \tag{5}$$

**Token Selecting.** For token selection, we return to the traditional Top-K selection policy for more controllable compression ratio, *i.e.*, preserve the Top-K important tokens while remove the other inattentive tokens, which is widely used in many works (Liang et al., 2022; Rao et al., 2021).

## 4.3 Token Pooling for Duplicative Redundancy

The token pooling techniques, also referred to as token merging or grouping , aims to merge similar image tokens together and can decrease the duplicative redundancy in the model. Recently, the Bipartite Soft Matching algorithm (BSM) algorithm (Bolya et al., 2023) shows superior performance in token merging. BSM first partitions the tokens into two sets of roughly equal size. It then draws an edge from each token in one set to the token in the other set with the highest *cosine* similarity score, where the tokens involved use the average across multiple heads to compute the similarity. The top-K-similar edges are selected, and tokens that are still connected are merged by averaging their features.

In addition, it is necessary to maintain a variable $\boldsymbol{s}$ that tracks the size of the tokens to minimize information loss. Formally, let $\boldsymbol{s} = [1, s_1, \dots, s_n] \in \mathbb{R}^{1 \times (n+1)}$ represent the size of each token, where $s_i$ is the number of original tokens that have been merged into the image token $i$, and $n$ is the number of reserved image tokens. This helps to preserve information by appropriately weighting each token during the attention calculation. In the multi-head attention mechanism, the attention matrix for each head is computed as follows:

$$\boldsymbol{A}^{(h)} = \text{Softmax}\left( \frac{\boldsymbol{Q}^{(h)} \cdot \boldsymbol{K}^{(h)T}}{\sqrt{d_h}} + \log \boldsymbol{s} \right), \tag{6}$$

where $\log \boldsymbol{s}$ is applied element-wise to adjust the attention scores based on the size of each token, ensuring that larger tokens (those representing more original tokens) are given greater importance. The outputs from all heads are then concatenated and linearly transformed to produce the final output, as described in the general MHSA formulation.

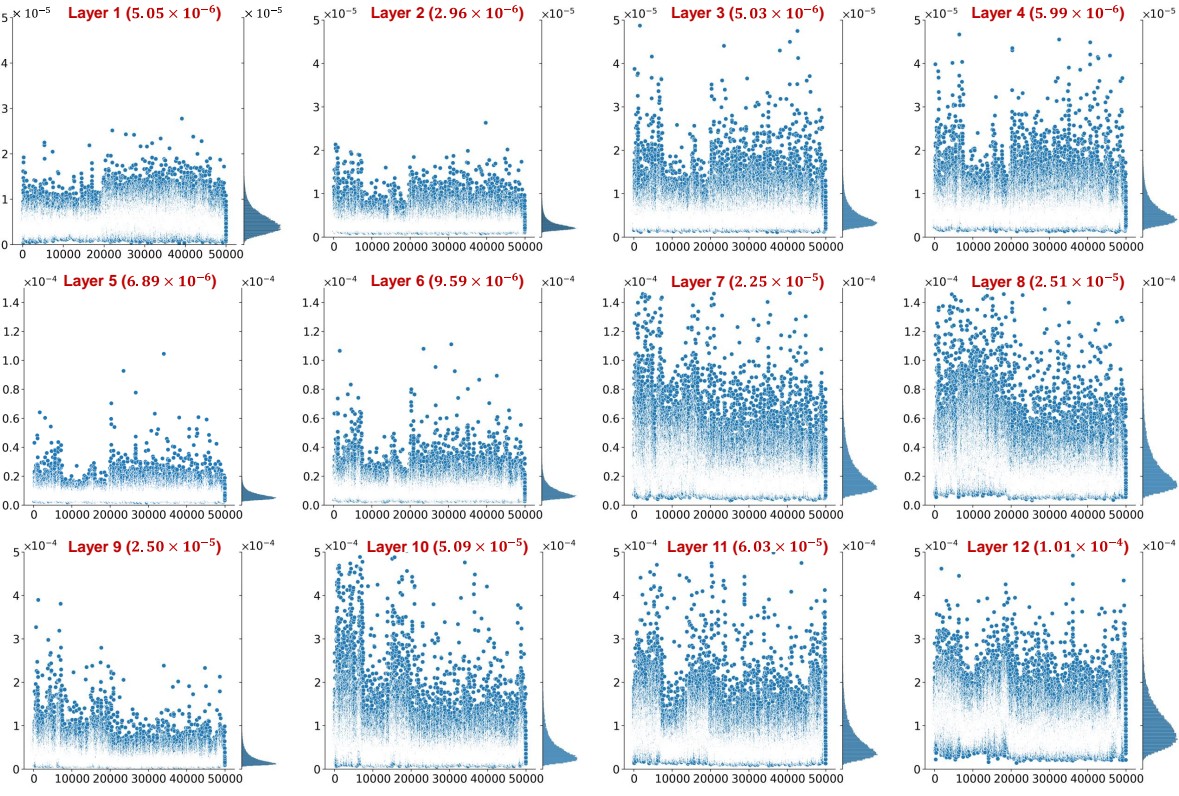

Figure 3: The scatter and the histogram of the variance of the significance scores assigned to image tokens at each layer of the DeiT-S model on the ImageNet validation set. The y-axis corresponds to the variance value and the x-axis to the index of samples in the dataset. We display the average variance of each layer at the top of each graph in red to track the trend of variance changes as the layers go deeper.

## 4.4 Token Pruning & Pooling Transformer

It is a non-trivial idea to combine token pruning for inattentive tokens and token pooling for attentive tokens. For example, an intuitive approach is to utilize both techniques within the same block. However, we find that this simple strategy does not obtain satisfactory performance, as shown in Section 5.2. To dig deeper into the possible reason behind, we first perform a comprehensive analysis of token pruning and token pooling techniques and find out an interesting observation for the token redundancy of different layers. To this end, we propose an adaptive token compression method to automatically discover the best policies to deal with the two types of redundancy in different layers.

**A Closer Look at Token Pruning and Token Pooling.** We first conduct some analysis of the importance scores in different layers. Our research reveals an intriguing phenomenon: the variance of significance scores assigned to image tokens for each sample increases as the number of layers in the model increases, as depicted in Figure 3. **(deeper analyses are shown in Section A of the Appendix)**. This suggests that *the importance of image tokens becomes more distinct as the layer deepens*. As a result, *token pruning techniques may be preferred at deeper layers*, where certain tokens exhibit significantly lower importance scores and are therefore more likely to be pruned. Meanwhile, premature token pruning at shallow layers may result in irreversible information loss and negatively impact model performance.

On the contrary, we find *that token pooling techniques are preferably applied in shallow layers* through our analysis. Since we use a pyramid compression approach for tokens, there are plenty of tokens in

the shallow layers that exhibit relatively high similarity. This makes it less concerning to merge dissimilar tokens during token pooling under the Top-K strategy, as it is unlikely to affect the performance of the model. Moreover, since we maintain the vector $\boldsymbol{s}$ that reflects the size of each token, the loss of information and the impact on model performance caused by merging highly similar tokens can be almost negligible. Additionally, due to the decreasing number of similar tokens, the utilization of token pooling techniques in deeper layers is not applicable.

**Adaptive Token Pruning & Pooling Strategy.** Based on the above discussion, we propose an adaptive token compression method to automatically discover the best policies for removing duplicative and inattentive redundancy. In this way, our method can adaptively select either token pruning **or** token pooling to tackle corresponding redundancy based on the current significance scores of tokens. In addition, we observe that the variance of the significance scores assigned to image tokens ***varies among different samples*** within the same layer in Figure 3. This implies that defining token compression rules solely based on the layer is not sufficient. Therefore, we propose an adaptive strategy that takes into account both the instances and layers, as follows:

$$S_{op_i} = var(\text{Score}_i), \ \text{Policy}_i = \begin{cases} \text{Token Pruning, if } S_{op_i} > \tau \\ \text{Token Pooling, otherwise} \end{cases}, \tag{7}$$

where the $\text{Score}_i$ is calculated in formula 5 and the $\tau$ is a hyperparameter, *i.e.*, decision threshold, that regulates the model's preference for either of the two strategies. The impact of $\tau$ on model performance is thoroughly explored in Section 5.2. Taking a holistic view of our approach, we find that *no additional learnable parameter* is introduced, which indicates that our method can be seamlessly integrated with pre-trained ViTs and yields competitive performance, as demonstrated by our experiment results.

**Adaptation to Multi-Stage Hierarchical Architectures.** Regarding the applicability to multi-stage hierarchical architectures of ViT variants, such as PvT and Swin, our approach can be adapted to these models with slight modifications. These models incorporate spatial reduction blocks within certain layers. For these blocks, we retain the attentive tokens from the entire token set at each compression stage and use masking or padding operations to preserve the full spatial structure during both training and inference. Formally, let $\boldsymbol{T}^{(l)} \in \mathbb{R}^{n_l \times d}$ represent the tokens at layer $l$, where $n_l$ is the number of tokens at that layer. During the token compression stage, we compute an attention score $\boldsymbol{A}^{(l)}$ for each token, and reserve the $k_l$ tokens based on our policy:

$$\boldsymbol{T}^{(l)}_{\text{reserved}} = \text{PPT}(\boldsymbol{A}^{(l)}, \boldsymbol{T}^{(l)}), \tag{8}$$

where $\boldsymbol{T}^{(l)}_{\text{reserved}} \in \mathbb{R}^{k_l \times d}$ represents the reserved tokens, and $k_l < n_l$ is the number of tokens retained. To preserve spatial consistency, we apply a masking or padding operation to the reserved tokens such that the spatial dimensions match the original structure:

$$\boldsymbol{T}^{(l)}_{\text{padded}} = \text{MaskPad}(\boldsymbol{T}^{(l)}_{\text{reserved}}, n_l), \tag{9}$$

where $\boldsymbol{T}^{(l)}_{\text{padded}} \in \mathbb{R}^{n_l \times d}$ is the token set after padding, ensuring that the spatial structure is maintained. This approach allows our method to be effectively integrated into multi-stage hierarchical architectures, preserving the spatial relationships crucial for dense prediction tasks.

## 5 Experiments

In this section, we empirically investigate the superiority of the proposed PPT through extensive experiments on ImageNet-1K (ILSVRC2012) (Deng et al., 2009), which contains approximately 1.28M training images and 50K validation images. We compare our proposed model with state-of-the-art models and conduct thorough ablation studies to gain a better understanding of its effectiveness.

**Implementation details.** We conduct experiments on the standard ViTs (Dosovitskiy et al., 2021) (including DeiT-Ti, DeiT-S, DeiT-B (Touvron et al., 2021)) and the different variants of ViTs (such as LV-ViT-S (Jiang et al., 2021), T2T-ViT(Yuan et al., 2021), and PS-ViT(Yue et al., 2021)). Following Liang et al. (2022), we specifically introduce our method into the $4^{th}$, $7^{th}$, and $10^{th}$ layers of DeiT models and into the $5^{th}$, $9^{th}$, and $13^{th}$ layers of LV-ViT models. For all comparative experiments, we report the performance of our method

Table 1: Comparison of accelerated vision transformers with various techniques on multiple standard ViTs on ImageNet-1K. '∗' denotes the incorporation of extra knowledge distillation. '†' indicates the models are trained from scratch. Our method achieves an improved balance between accuracy and computational efficiency, requiring fewer fine-tuning epochs. Moreover, when employed in an off-the-shelf (without fine-tuning) setting, it delivers comparable performance to other fine-tuned models.

| Model | Method | Epochs | Top-1 ACC (%) | Params (M) | FLOPs (G) | Throughput (image/s) |
|---|---|---|---|---|---|---|
| DeiT-Ti | Baseline | - | 72.2 | 5.6 | 1.3 | 2675 |
| | DynamicViT∗ | 30 | 71.4 (↓ 0.8) | 5.9 | 0.8 (↓ 38.5%) | 3765 (↑ 40.7%) |
| | Evo-ViT† | 300 | 72.0 (↓ 0.2) | 5.9 | 0.8 (↓ 38.5%) | 3781 (↑ 41.3%) |
| | EViT | 30 | 71.9 (↓ 0.3) | 5.6 | 0.8 (↓ 38.5%) | 3387 (↑ 26.6%) |
| | ToMe† | 300 | 71.4 (↓ 0.8) | 5.6 | 0.8 (↓ 38.5%) | 3685 (↑ 37.7%) |
| | **PPT (Ours) (Off-the-shelf)** | 0 | 71.6 (↓ 0.6) | 5.6 | 0.8 (↓ 38.5%) | 3572 (↑ 33.5%) |
| | **PPT (Ours)** | 30 | **72.1 (↓ 0.1)** | 5.6 | 0.8 (↓ 38.5%) | 3572 (↑ 33.5%) |
| DeiT-S | Baseline | - | 79.8 | 22.1 | 4.6 | 993 |
| | DynamicViT∗ | 30 | 79.3 (↓ 0.5) | 22.8 | 3.0 (↓ 34.8%) | 1440 (↑ 45.0%) |
| | IA-RED² | 90 | 79.1 (↓ 0.7) | - | 3.2 (↓ 30.4%) | - |
| | PS-ViT | 30 | 79.4 (↓ 0.4) | 22.1 | 2.6 (↓ 43.5%) | 1321 (↑ 33.0%) |
| | Evo-ViT† | 300 | 79.4 (↓ 0.4) | 22.4 | 3.0 (↓ 34.8%) | 1414 (↑ 42.4%) |
| | EViT | 30 | 79.5 (↓ 0.3) | 22.1 | 3.0 (↓ 34.8%) | 1378 (↑ 38.8%) |
| | ATS | 30 | 79.7 (↓ 0.1) | 22.1 | 2.9 (↓ 37.0%) | 1382 (↑ 39.2%) |
| | ToMe† | 300 | 79.4 (↓ 0.4) | 22.1 | 2.7 (↓ 41.3%) | 1552 (↑ 56.3%) |
| | TPS | 30 | 79.7 (↓ 0.1) | 22.1 | 3.0 (↓ 34.8%) | 1392 (↑ 40.2%) |
| | SViT | 30 | 79.4 (↓ 0.4) | 22.1 | 3.0 (↓ 34.8%) | - |
| | **PPT (Ours) (Off-the-shelf)** | 0 | 79.5 (↓ 0.3) | 22.1 | 2.9 (↓ 37.0%) | 1448 (↑ 45.8%) |
| | **PPT (Ours)** | 30 | **79.8 (↓ 0.0)** | 22.1 | 2.9 (↓ 37.0%) | 1448 (↑ 45.8%) |
| DeiT-B | Baseline | - | 81.8 | 86.6 | 17.6 | 295 |
| | DynamicViT | 30 | 81.3 (↓ 0.4) | 89.4 | 11.5 (↓ 34.6%) | 454 (↑ 53.8%) |
| | IA-RED² | 90 | 80.3 (↓ 1.5) | - | 11.8 (↓ 33.0%) | 453 (↑ 53.6%) |
| | Evo-ViT† | 300 | 81.3 (↓ 0.5) | 87.3 | 11.7 (↓ 33.5%) | 429 (↑ 45.4%) |
| | EViT | 30 | 81.3 (↓ 0.5) | 86.6 | 11.6 (↓ 34.1%) | 440 (↑ 49.2%) |
| | **PPT (Ours) (Off-the-shelf)** | 0 | 80.3 (↓ 1.5) | 86.6 | 11.6 (↓ 34.1%) | 445 (↑ 50.8%) |
| | **PPT (Ours)** | 30 | **81.4 (↓ 0.4)** | 86.6 | 11.6 (↓ 34.1%) | 445 (↑ 50.8%) |

in terms of both *off-the-shelf* and *fine-tuned*. Following the approach in Rao et al. (2021), we initialize the backbone models with official pre-trained ViTs. If fine-tuning is applied, we jointly train the entire models for 30 epochs, similar to other works (Fayyaz et al., 2022; Liang et al., 2022; Rao et al., 2021). Regarding training strategies and optimization methods, we follow the setup described in the original papers of DeiT and LV-ViT, except that the basic learning rates are set to $\frac{batchsize}{128} \times 10^{-5}$. The image resolution used for both training and testing is $224 \times 224$. The decision threshold $\tau$ we introduced in our framework is $7 \times 10^{-5}$ and $5 \times 10^{-4}$ in DeiT and LV-ViT-S, respectively. All experiments are conducted using PyTorch on NVIDIA GPUs. We report the classification accuracy of the top-1 and floating-point operations (FLOPs) to evaluate the efficiency of the model. Additionally, we measure the throughput of the models on a single NVIDIA V100 GPU with batch size fixed to 256 same as Tang et al. (2022); Xu et al. (2022).

## 5.1 Main Results

**Comparisons with existing methods.** Even though we do not add extra parameters as Rao et al. (2021), or apply complicated token reorganization tricks as Liang et al. (2022); Xu et al. (2022), the experimental results, as presented in Table 1 and Figure 4, demonstrate that our method achieves higher accuracy with comparable computation cost. Specifically, PPT reduces FLOPs by over 37% and improves throughput by over 45% without any drop in accuracy in DeiT-S, as shown in Table 1. Furthermore, the superiority of PPT is evident across various FLOPs compared to other token compression methods, as shown in Figure 4.

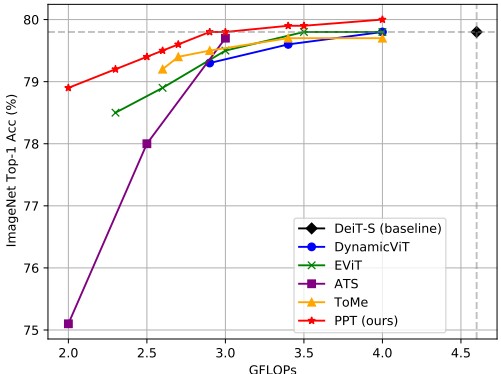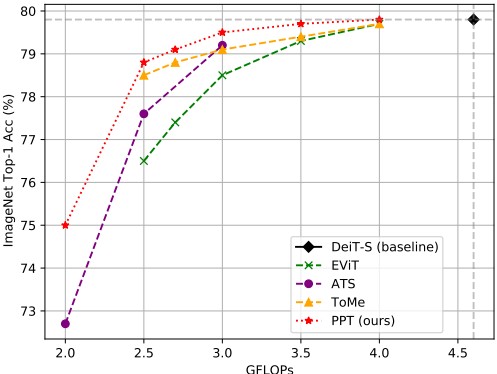

Figure 4: **Comparison between our method and other methods under different FLOPs**. We conducted a comprehensive comparison of the performance of various methods after **fine-tuning (left)** and **off-the-shelf (right)**, which highlights the superiority of our method.

Specifically, our method excels in the following three aspects: First, at lower FLOPs (below 2.5 G), our method shows a notable performance improvement of 0.7%-3.8% compared to other methods. Second, at higher FLOPs (3.0 G above), our method outperforms the baseline model and surpasses the capabilities of the comparison methods. Lastly, when used in an off-the-shelf setting, PPT shows a significant improvement of 0.1%-2.3% across various FLOPs compared to other methods, achieving competitive results with those obtained by fine-tuned models.

**Application on different variants of ViTs.** Apart from the standard ViTs, subsequent studies (Chen et al., 2021; Han et al., 2021; Heo et al., 2021; Jiang et al., 2021; Liu et al., 2021; Radosavovic et al., 2020; Wang et al., 2021; Wu et al., 2021; Xu et al., 2021; Yuan et al., 2021; Yue et al., 2021) further improve the performance of ViTs by varying the initial architecture or optimization strategies. To further showcase the potential and superiority of our approach, we extend the integration of PPT with the LV-ViT-S. As shown in Table 2 and Figure 5, our PPT-LV-S can achieve a highly competitive performance among numerous vision transformers from the perspective of accuracy-computation trade-off, and further demonstrate superiority over other token compression methods on LV-ViT-S. To further demonstrate the generality of our approach, we also apply PPT on T2T-ViT and PS-ViT, as shown in top 2 rows of Figure 6. Additionally, we validate the applicability of our method to multi-stage hierarchical architectures, such as PvT and Swin, with experimental results under the *off-the-shelf* configuration, as illustrated in the last row of Figure 6.

**Video experiments.** We also extended our experiments to video classification, a promising direction for applying our PPT combined with Spatiotemporal MAE (Feichtenhofer et al., 2022) on the Kinetics-400 (Kay et al., 2017) dataset. In the Table 3, we present the results of applying our method both off-the-shelf and during MAE fine-tuning using ViT-L from Spatiotemporal MAE compared to the relevant state-of-the-art on Kinetics-400 classification across different FLOPs levels. The comparisons include Swin (Liu et al., 2021) pretrained on ImageNet-21k, MViTv2 Li et al. (2022b) pretrained with MaskFeats (Wei et al., 2022), and Spatiotemporal MAE as the baseline. For completeness, we also include a token pruning work, X-ViT+ATS (Fayyaz et al., 2022), and a token pooling work, ToMe (Bolya et al., 2023). These results show our approach achieves a better balance between speed and accuracy compared to the state-of-the-art.

**Visualizations of token compression results.** To gain further insight into the interpretability of PPT, we conducted a visualization analysis of the intermediate process of our token compression in Figure 7. As expected, we observe that PPT tends to implement token pooling in shallow layers and token pruning in deep layers, and leverage incompletely consistent compression strategies for different images. As the network deepens, the duplicative redundancy and inattentive redundancy gradually removed, while the most informative tokens are reserved. From the final output, we observed that PPT can assist ViTs in focusing on patches specific to the target class, and we also demonstrated that background patches can be meaningful for recognition. The visualizations corroborate that our approach is effective in processing images irrespective of whether the backgrounds are simple or complex. And the results demonstrate that the model is more cautious when processing complex and information-rich images, preferring token pooling strategies with lower

Table 2: Comparisons with different variants of ViTs on ImageNet. We compress the LV-ViT-S (Jiang et al., 2021) as the base model and achieve promising accuracy-FLOPs trade-off.

| Model | Params (M) | FLOPs (G) | Top-1 Acc. (%) |
|---|---|---|---|
| DeiT-S | 22.1 | 4.6 | 79.8 |
| DeiT-B | 86.6 | 17.6 | 81.8 |
| PVT-Small | 24.5 | 3.8 | 79.8 |
| PVT-Medium | 44.2 | 6.7 | 81.2 |
| CoaT-Lite Small | 20.0 | 4.0 | 81.9 |
| CrossViT-S | 26.7 | 5.6 | 81.0 |
| Swin-T | 29.0 | 4.5 | 81.3 |
| Swin-S | 50.0 | 8.7 | 83.0 |
| T2T-ViT-14 | 22.0 | 4.8 | 81.5 |
| T2T-ViT-24 | 64.1 | 14.1 | 82.3 |
| RegNetY-8G | 39.0 | 8.0 | 81.7 |
| RegNetY-16G | 84.0 | 16.0 | 82.9 |
| PS-ViT-B/14 | 21.3 | 5.4 | 81.5 |
| PS-ViT-B/18 | 21.3 | 8.8 | 82.3 |
| PiT-S | 23.5 | 2.9 | 80.9 |
| PiT-B | 73.8 | 12.5 | 82.0 |
| CvT-13 | 20.0 | 4.5 | 81.6 |
| TNT-S | 23.8 | 5.2 | 81.5 |
| TNT-B | 66.0 | 14.1 | 82.9 |
| LV-ViT-S | 26.2 | 6.6 | 83.3 |
| DynamicViT-LV-S | 26.9 | 4.6 | 83.0 |
| PS-LV-ViT-S | 26.2 | 4.7 | 82.4 |
| EViT-LV-S | 26.2 | 4.7 | 83.0 |
| **PPT-LV-S (Ours) (off-the-shelf)** | 25.8 | 4.6 | 82.8 |
| **PPT-LV-S (Ours)** | **25.8** | **4.6** | **83.1** |

Table 3: Results for our method without training or applied during MAE fine-tuning compared to SoTA on Kinetics-400 in the same flop range.

| Model | Input | Acc(%) | GFLOPs |
|---|---|---|---|
| ViT-B (MAE) | $16\times224^2$ | 81.3 | $180 \times 3 \times 7$ |
| Swin-B | $32\times224^2$ | 82.7 | $282 \times 3 \times 4$ |
| XViT + ATS | $16\times224^2$ | 80.0 | $259 \times 1 \times 3$ |
| ToMe + ViT-L (MAE) | $16\times224^2$ | 83.2 | $184 \times 1 \times 10$ |
| **PPT + ViT-L (MAE)** | | | |
| *Off-the-shelf* | $16\times224^2$ | 82.8 | $180 \times 1 \times 10$ |
| *Fine-tuning* | $16\times224^2$ | **83.5** | $\mathbf{180 \times 1 \times 10}$ |
| ViT-L (MAE) | $16\times224^2$ | 84.7 | $598 \times 1 \times 10$ |
| Swin-L | $32\times224^2$ | 83.1 | $604 \times 3 \times 4$ |
| MViTv2-L | $16\times224^2$ | 84.3 | $377 \times 1 \times 10$ |
| ToMe + ViT-L (MAE) | $16\times224^2$ | 84.5 | $281 \times 1 \times 10$ |
| **PPT + ViT-L (MAE)** | | | |
| *Off-the-shelf* | $16\times224^2$ | 84.6 | $270 \times 1 \times 10$ |
| *Fine-tuning* | $16\times224^2$ | **84.8** | $\mathbf{270 \times 1 \times 10}$ |

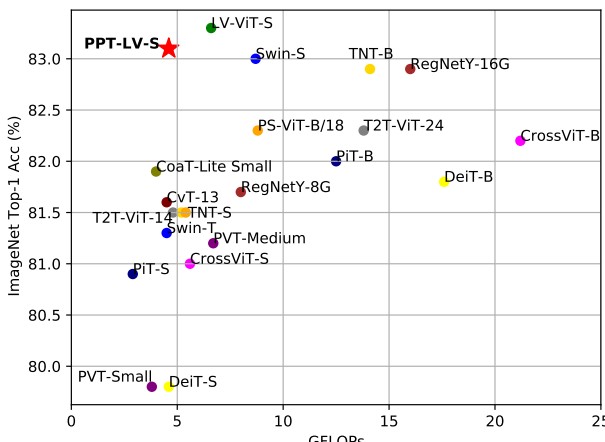

Figure 5: Comparison of different models with various accuracy-FLOPs trade-off. Our PPT-LV-S achieves a quite competitive trade-off than other variants of ViTs.

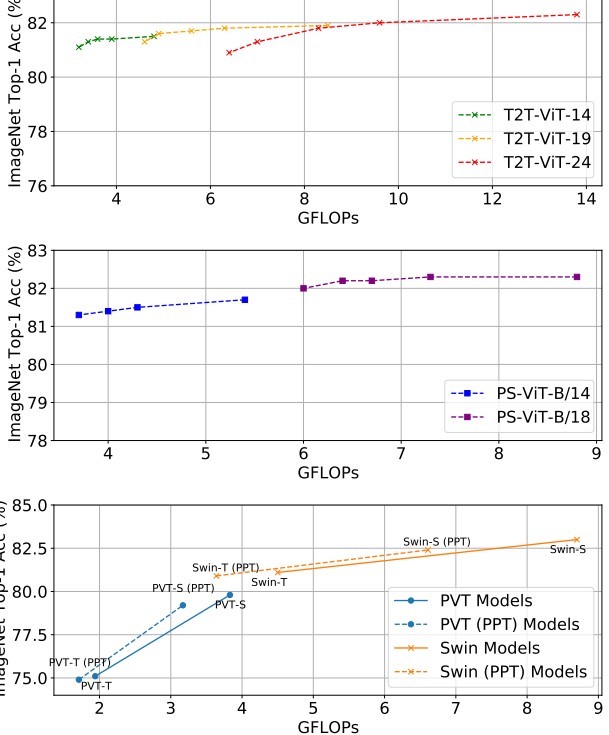

Figure 6: The performance of applying our method to more variants of ViTs (without fine-tuning).

information loss. Conversely, for simpler images, the model is more decisive and tends to use token pruning strategies. These observations intuitively underscore the importance of our adaptive token compression strategy to a certain extent.

## 5.2 Ablation Study

We conduct extensive ablation studies to explore the effectiveness of each component in our method. The DeiT-S is used as the default model and ImageNet-1k as the default dataset.

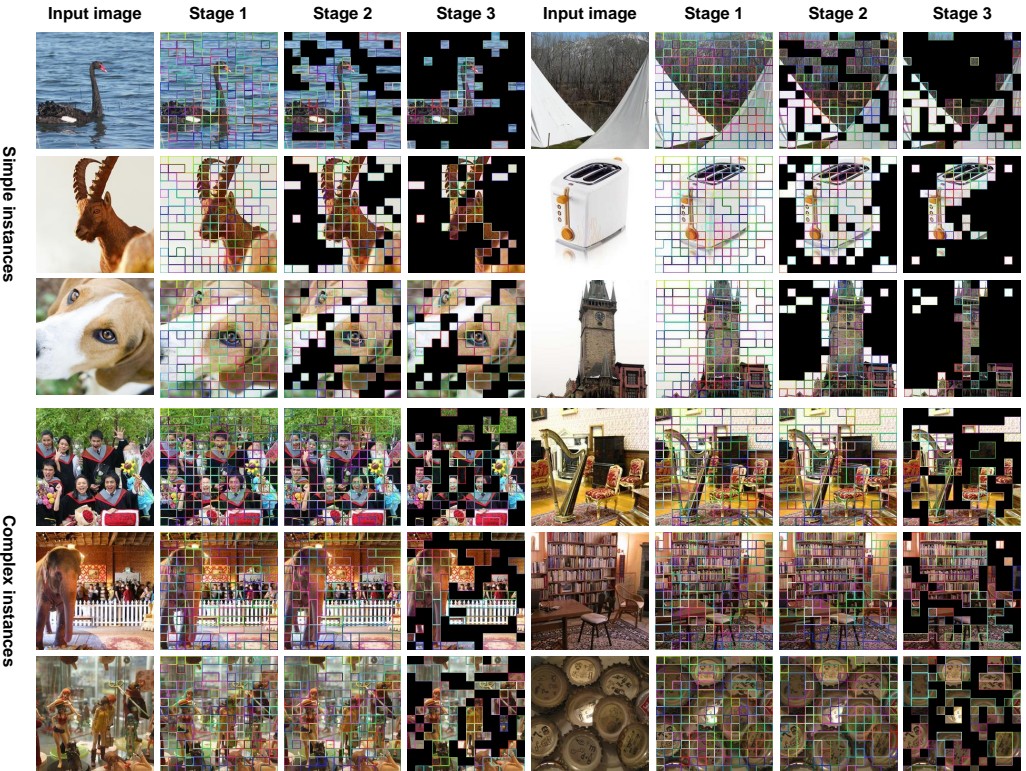

Figure 7: **Visualizations of token compression results on DeiT-S**. The masked regions represent the inattentive redundancy and are pruned, whereas the patches with the same inner and border color mean the duplicative redundancy and are pooled. The redundancy in the image is pyramidally reduced, and our method can adaptively execute different strategies at different stages based on the input. We demonstrate the generalization of our method to images of varying complexity. More results are visualized in Section C of the appendix.

**Effectiveness of techniques integration.** We evaluate the effectiveness of proposed framework by examining the performance of *token pruning*, *token pooling*, and *naively combining pruning and pooling within a block* - in different FLOP settings. As shown in Figure 8, token pooling outperforms token pruning when used independently *in our framework*. The naively combining demonstrates an advantage at lower FLOPs but can diminish the benefits of token pooling at higher FLOPs. In contrast, our framework consistently achieves optimal performance across different FLOPs, highlighting its effectiveness.

**Impact of the decision threshold.** As a key hyperparameter in PPT, the decision threshold $\tau$ controls the model's preference for token pruning and token pooling techniques. When the $\tau$ is a smaller value, the model is more likely to use the token pruning policy, and vice versa. It is important to set an optimal $\tau$ to

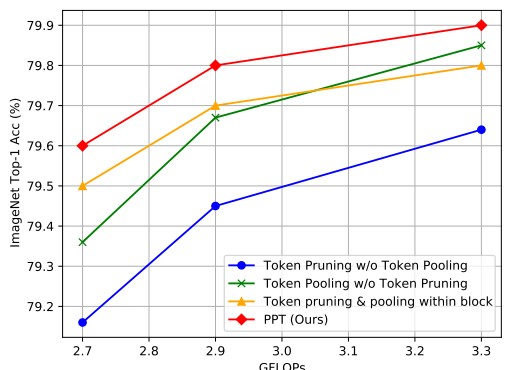

Figure 8: Comparison of the PPT framework with the individual modules and an alternative combination method.

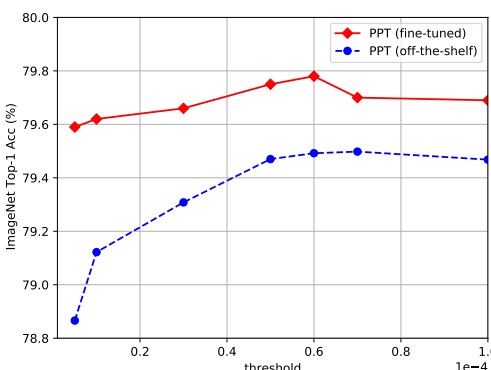

Figure 9: Impact of the decision threshold $\tau$ on the performance with or without training.

balance the two strategies. Motivated by the statistical data in Figure 3, we explore the $\tau$ range from the average variance of first layer ($5 \times 10^{-6}$) to the last layer ($1 \times 10^{-4}$) and show the results in Figure 9. We observe that the optimal $\tau$ is $6 \times 10^{-5}$ with fine-tuning and $7 \times 10^{-5}$ under *on-the-shelf*, respectively.

**Different policy selection mechanisms.** We believe that different images possess varying levels of complexity and exhibit different types of redundancy. Therefore, dynamically selecting strategies based on input can enhance the flexibility of the model, as demonstrated in Figure 3 and Figure 7. To further demonstrate the necessity of this strategy, we conduct additional experiments (under an off-the-shelf scenario) to compare our adaptive selection strategy with the random policy and the rule-based policy. The rule-based policy involved using token pooling for the initial half of the blocks and transitioning to token pruning for the remaining ones. Additionally, we explored the inversion of our policy decision, wherein pruning was performed instead of pooling, and vice versa. This exploration aimed to reinforce our claims that token pooling techniques are typically applied in shallow layers, while token pruning techniques are preferably employed in deeper layers. Detailed experimental results are presented in Table 4.

Table 4: Effectiveness of different policy selection mechanism.

| FLOPs (G) | Top-1 Acc(%) of various mechanisms | | | |
|---|---|---|---|---|
| | Random | Rule-Based | **Adaptive (Ours)** | Policy Inversion |
| 2.5 | 77.5 | 78.4 | **78.8** | 76.5 |
| 2.7 | 78.2 | 78.8 | **79.1** | 77.6 |
| 2.9 | 79.0 | 79.2 | **79.5** | 78.6 |

**Different metrics for redundancy discrimination.** The reason we use the variance of the significant score as our policy score in our approach is because the variance reflects the dispersion of the values. A larger variance indicates a clearer distinction between important and unimportant image tokens, which is beneficial for token pruning. Conversely, when the variance is smaller, token pooling is more suitable. Additionally, we have explored another metric, the average of token similarity, which also reflects the redundancy level of tokens in different layers. However, through ablation study (without fine-tuning), we have observed that its performance is inferior to our chosen metric.

Table 5: Comparison between different metrics for redundancy discrimination.

| Metrics | GFLOPs | Top-1 ACC (%) |
|---|---|---|
| average of token similarity | 2.9 | 79.2 |
| **variance of the significant score** | | 79.5 |

## 6 Conclusion

In this work, we bring a new perspective for obtaining efficient vision transformers by integrating both token pruning and token pooling techniques. Our proposed framework, named token Pruning & Pooling Transformers (PPT), adaptively decides different token compression policies for various layers and instances. The proposed method is simple, yet effective, and can be easily incorporated into the standard transformer block without additional trainable parameters. Extensive experiments demonstrate the effectiveness of our method. Specifically, our PPT can reduce more than 37% FLOPs and improve the throughput by over 45% for DeiT-S without any reduction in accuracy on ImageNet. While our work primarily focuses on image classification tasks, we see the potential to extend our approach to dense prediction tasks, such as semantic segmentation and object detection. Although token pruning may lead to some spatial information loss, we believe that careful adaptations—such as selectively retaining key tokens and using efficient upsampling techniques—could help address these challenges. We plan to explore these possibilities in future work to further extend the applicability of our method.

Overall, our work offers a valuable contribution to the development of efficient vision transformers, highlighting the importance of adaptive token compression and providing new insights into the integration of token pruning and token pooling techniques. We hope that our method inspires new research and leads to further improvements in the field of efficient transformer-based models.

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

## A    Deeper Analysis

**Distributions of significance scores.** In Figure 10, we further investigate the impact of our method on the distribution of significance scores across different layers. Specifically, we introduce our method into the $4^{th}$, $7^{th}$, and $10^{th}$ layers of the DeiT-S model. As shown in comparison to the original model (Figure 3 in the main text), the first four layers remain unaffected, while the introduction of PPT in the fourth layer leads to a significant increase in variance for the subsequent layers. This phenomenon indicates that ***our method makes the importance of image tokens more distinct***, which may be the underlying reason for the performance improvement achieved by our method.

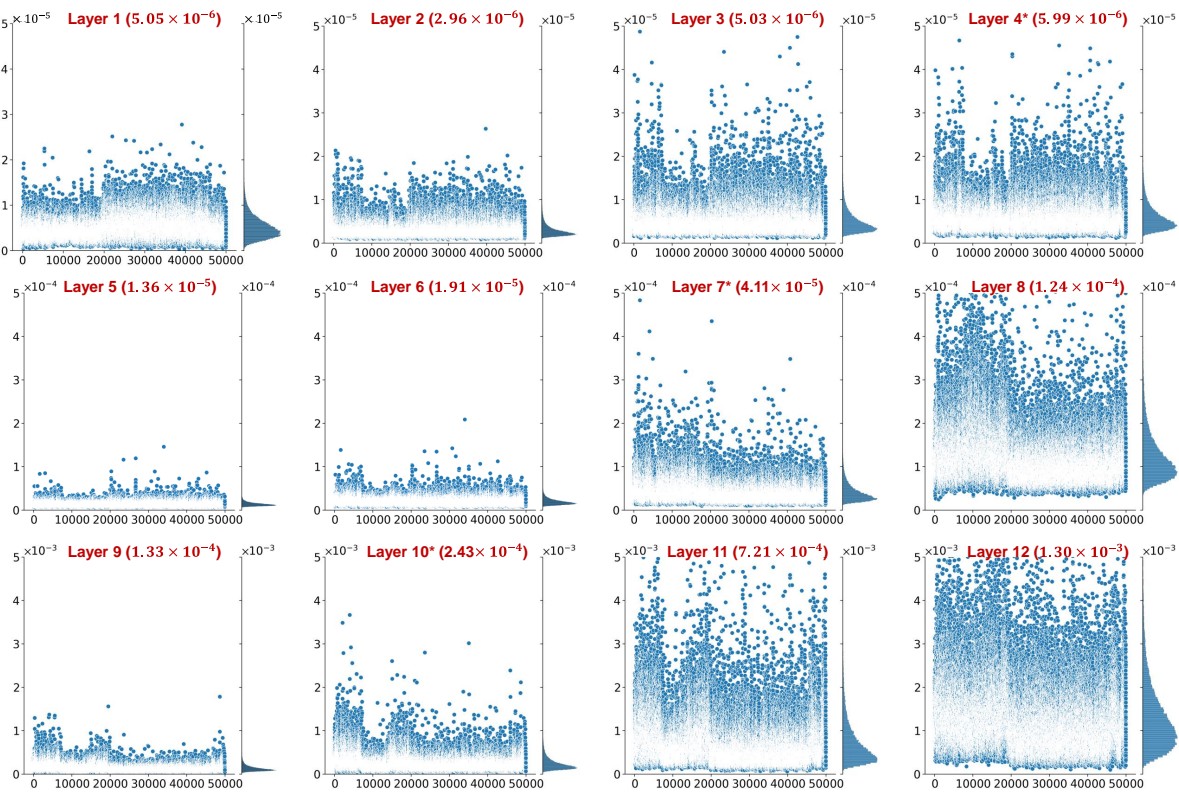

Figure 10: The scatter and the histogram of the variance of the significance scores assigned to image tokens at each layer of the **compressed** DeiT-S model on the ImageNet validation set, t**he layer with '*' indicate the PPT block is inserted**. The y-axis corresponds to the variance value and the x-axis to the index of samples in the dataset. We display the average variance of each layer at the top of each graph.

In Figure 11, we further demonstrate our analysis by computing the average variance of significance scores across layers for different ViTs. We can observe a similar phenomenon across different ViTs, where ***the variance increases with layer depth, and compressed ViTs exhibit greater variance.*** This phenomenon to some extent validates the effectiveness and generality of our method.

## B    Full Results

We provide more detailed performance results of PPT on DeiT and LV-ViT-S in Table 6. Again, the superiority of PPT is evident under different FLOPs. Specifically, PPT maintains remarkably high accuracy even at higher compression rates, and interestingly, it even outperforms the original model at lower compression rates.

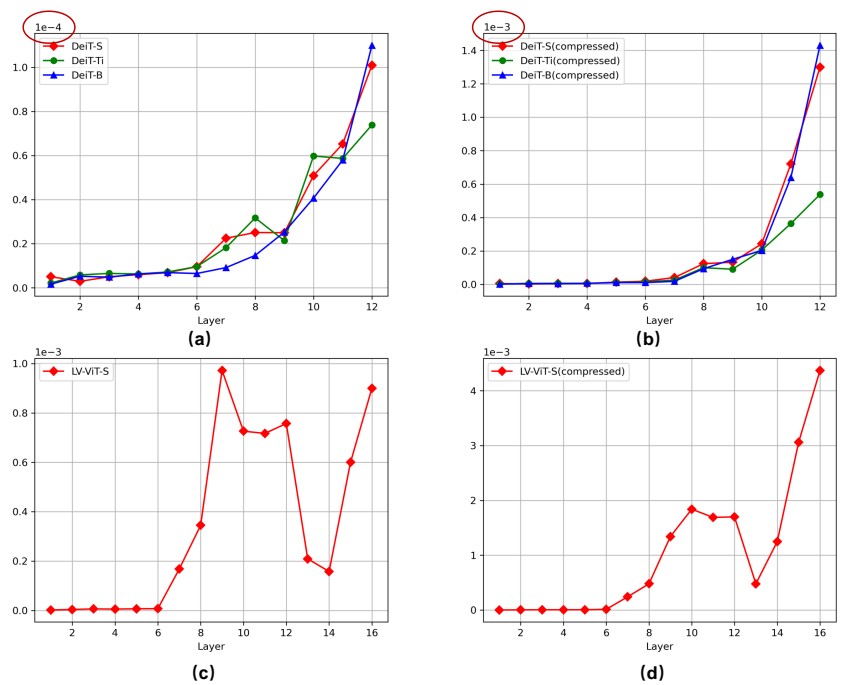

Figure 11: **The average variance of the significance scores at each layer** of different ViTs on the ImageNet validation set. Figure (a) and Figure (c) are generated based on the original models, while Figure (b) and Figure (d) are generated based on the compressed models with our method.

Table 6: Results of PPT on different ViTs under various FLOPs, ACC* indicates that the ACC is evaluated under off-the-shelf (without fine-tuning).

| Model | Removed tokens per stage | FLOPs (G) | Top-1 Acc. (%) | Top-1 Acc*. (%) |
|---|---|---|---|---|
| ViT (DeiT)-Ti | 0 | 1.3 | 72.2 | 72.2 |
| | 10 | 1.16 | 72.32 | 72.09 |
| | 20 | 1.07 | 72.26 | 72.05 |
| | 30 | 0.97 | 72.25 | 72.06 |
| | 40 | 0.89 | 72.10 | 71.74 |
| | 45 | 0.84 | 71.99 | 71.61 |
| | 50 | 0.80 | 71.90 | 71.28 |
| | 60 | 0.74 | 71.58 | 70.54 |
| ViT (DeiT)-S | 0 | 4.6 | 79.8 | 79.8 |
| | 10 | 4.26 | 79.99 | 79.79 |
| | 20 | 3.92 | 80.00 | 79.81 |
| | 30 | 3.59 | 79.94 | 79.72 |
| | 40 | 3.26 | 79.89 | 79.65 |
| | 50 | 2.94 | 79.76 | 79.49 |
| | 60 | 2.72 | 79.58 | 79.13 |
| ViT (DeiT)-B | 0 | 17.6 | 81.8 | 81.8 |
| | 40 | 12.48 | 81.47 | 80.88 |
| | 47 | 11.60 | 81.37 | 80.30 |
| | 50 | 11.27 | 81.19 | 80.04 |
| LV-ViT-S | 0 | 6.6 | 83.3 | 83.3 |
| | 40 | 4.94 | 83.25 | 83.03 |
| | 50 | 4.60 | 83.09 | 82.82 |
| | 60 | 4.33 | 82.82 | 82.57 |

In addition, we provide the detailed data for plots in Figure 4, as shown in Table 7 and Table 8.

Table 7: The corresponding data for Figure.4 (a)

| FLOPs (G) | Top-1 Acc(%) of Models (Fine-tunned) | | | | |
|---|---|---|---|---|---|
| | Dynamic ViT | EViT | ATS | ToMe | **PPT (Ours)** |
| 2.0 | - | - | 75.1 | - | **78.9** |
| 2.3 | - | 78.5 | - | - | **79.2** |
| 2.5 | - | - | 78.0 | - | **79.4** |
| 2.6 | - | 78.9 | - | 79.2 | **79.5** |
| 2.7 | - | - | - | 79.4 | **79.6** |
| 2.9 | 79.3 | - | - | 79.5 | **79.8** |
| 3.0 | - | 79.5 | 79.7 | - | **79.8** |
| 3.4 | 79.6 | - | - | 79.7 | **79.9** |
| 3.5 | - | 79.8 | - | - | **79.9** |
| 4.0 | 79.8 | 79.8 | - | 79.7 | **80.0** |

Table 8: The corresponding data for Figure.4 (b)

| FLOPs (G) | Top-1 Acc(%) of Models (Off-the-shelf) | | | |
|---|---|---|---|---|
| | EViT | ATS | ToMe | **PPT(Ours)** |
| 2.0 | - | 72.7 | - | **75.0** |
| 2.5 | 76.5 | 77.6 | - | **78.8** |
| 2.7 | 77.4 | - | 78.8 | **79.1** |
| 3.0 | 78.5 | 79.2 | 79.1 | **79.5** |
| 3.5 | 79.3 | - | 79.4 | **79.7** |
| 4.0 | 79.7 | - | 79.7 | **79.8** |

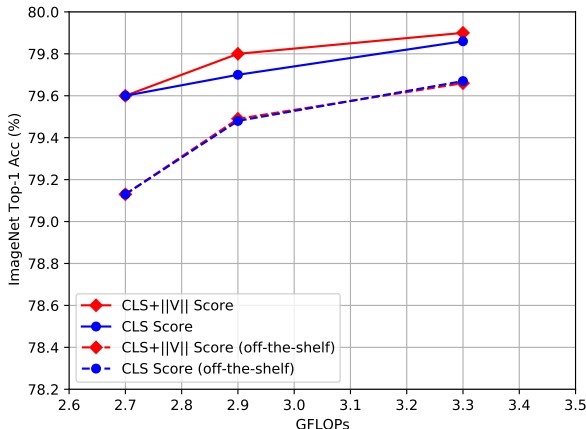 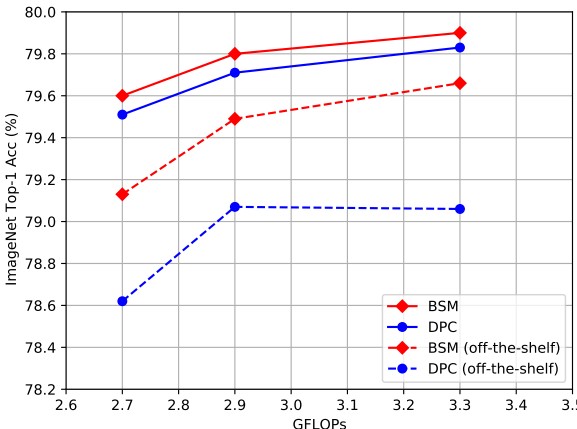

Figure 12: Impact of different score assignment methods when token pruning applied in our framework.

Figure 13: BSM *V.S.* DPC when token pooling applied in our framework

**Different token pooling policy.** Here we discuss different design choices for token pooling. For example, we could also utilize the efficient density peak clustering algorithm (DPC) (Rodriguez & Laio, 2014) for token merging. It identifies density peaks based on the local density and distance between image tokens, and assigns each token to the nearest density peak and forms clusters by merging nearby tokens that belong to the same peak. Figure 13 presents the effects of different token pooling policies. Our experiments demonstrate that BSM can surpass the results of DPC by approximately 0.1% with fine-tuning, and the advantage gap is even larger without training.

**Different token pruning policy.** As described in Section 4.2, considering the values matrix $V$ plays a critical role in influencing the model's performance. To delve deeper, we conducted a comprehensive exploration of various token scoring mechanisms. As illustrated in Figure 12, we observed that the performance of the two methods is closely comparable when evaluated *off-the-shelf*. However, scoring with the norm of $V$ yields slightly better results after fine-tuning. This indicates that incorporating the norm of $V$ can enhance the model's effectiveness, especially after the fine-tuning process. These insights highlight the nuanced impact of different scoring mechanisms on model performance.

## C More Visualization

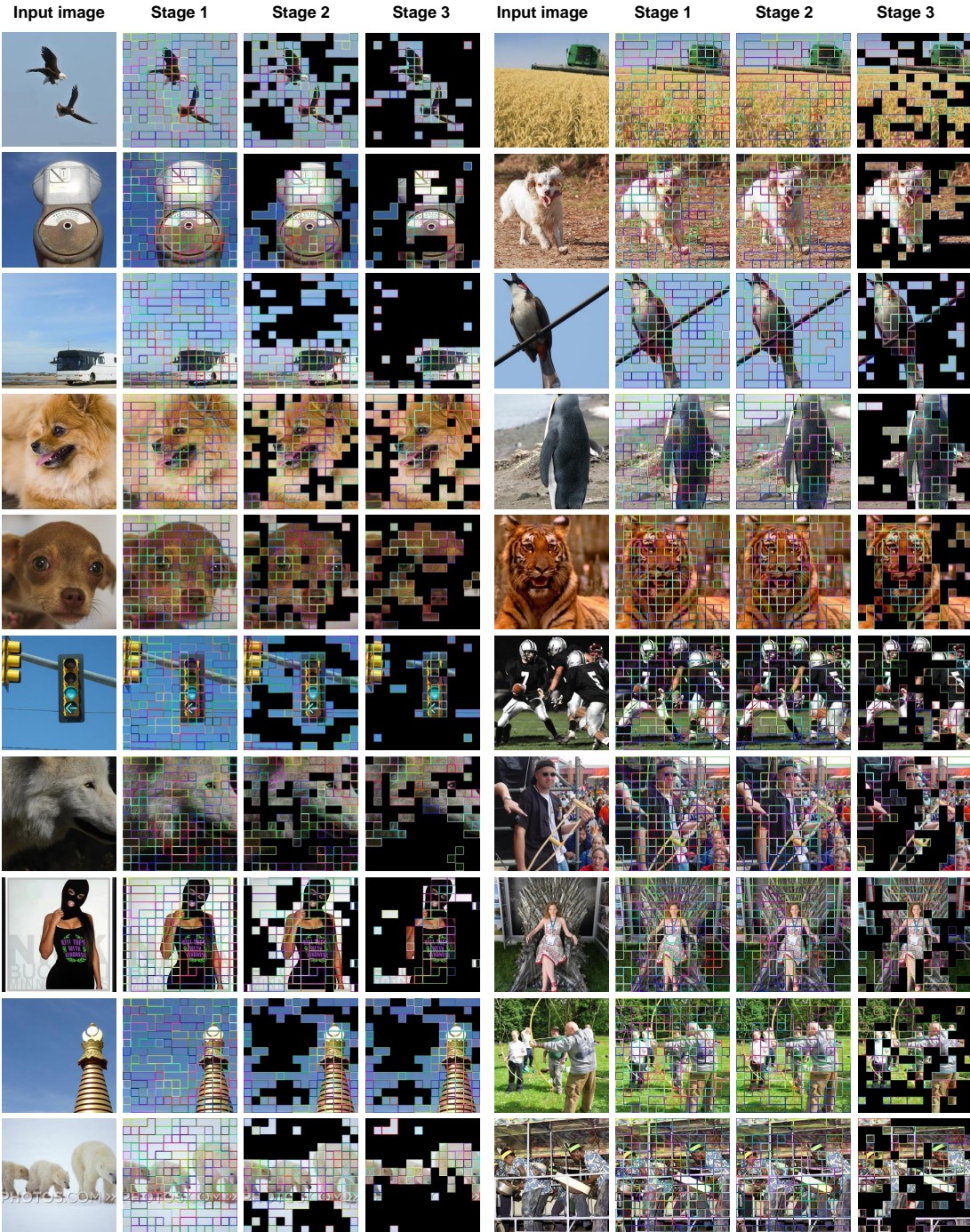

Figure 14: **Extended visualizations of our token compression results on DeiT-S with 12 layers**. The input image is sampled from the validation of ImageNet. The masked regions represent the inattentive redundancy and are pruned, while the patches with the same inner and border color means the duplicative redundancy and are pooled. The redundancy in the image is pyramid reduced, and our method can adaptively execute different strategies at different stages based on the input. We demonstrate our method works well for different images from various categories with various complexity.

