# OpenReview forum: "PPT: Adaptive Token Pruning and Pooling for Efficient Vision Transformers"
_TMLR — Rejected by TMLR_

### Review · Reviewer_ACnx · 2024-08-19

**Summary Of Contributions:**

This paper proposed an efficient pruning and pooling method for ViT architectures called PPT -- The approach adaptively apply token pruning and token pooling during inference using a criterion. The approach is simple yet effective. Also the approach is training free. The approach could strike a balance between FLOPs and accuracy on imagenet dataset.

**Audience:**

Yes

**Claims And Evidence:**

Yes

**Requested Changes:**

The paper is technically sound but I think addressing the weakness sections would be helpful for improving the future impact of this approach.

**Strengths And Weaknesses:**

[Strength]
- The approach is simple yet seemingly effective. The approach is also training-free so the actual productization of the approach may be simple
- The method combined token pruning and token pooling in an interesting manner. I would say that there are some novelties despite the approach is fairly simple

[Weaknesses]
- Because the approach is simple yet effective on imagenet dataset, I was expecting to see more results on 1) more datasets and 2) more CV tasks beyond image classification
- The paper only reported FLOPs but not actual run time/latency on real hardware. This may hinder the claims for the effectiveness of the approach
- Most of SOTA vision transformers are now being or will be replaced by Swin-like approaches, I wonder if the authors could discuss if the approach could apply to those architectures

---

> ### Author Response · Authors · 2024-09-02
> **Response to Reviewer ACnx**
>
> We sincerely appreciate your feedback. We respond to each of your concerns and suggestions one-by-one in what follows:
>
> > **More Results on Additional Datasets and Tasks**
>
> Thank you for your insightful comments. We have extended our experiments to **video classification**, a promising direction for applying our PPT combined with [**Spatiotemporal MAE** (Feichtenhofer et al., 2022)](https://arxiv.org/abs/2205.09113) on the [**Kinetics-400** (Kay et al., 2017)](https://arxiv.org/abs/1705.06950) dataset. **In the table 3 of our newly upload paper**,  we present the results of applying our method both off-the-shelf and during MAE fine-tuning using ViT-L from Spatiotemporal MAE compared to the relevant state-of-the-art on Kinetics-400 classification across different FLOPs levels. The comparisons include [Swin (Liu et al., 2022)](https://arxiv.org/abs/2106.13230) pretrained on ImageNet-21k, [MViTv2 (Li et al.,2022)](https://arxiv.org/abs/2112.01526) pretrained with [MaskFeats (Wei et al., 2022)](https://arxiv.org/abs/2112.09133), and Spatiotemporal MAE as the *baseline*. For completeness, we also include a token pruning work, [X-ViT + ATS (Fayyaz et al., 2022)](https://arxiv.org/abs/2111.15667), and a token pooling work, [ToMe (Bolya et al., 2023)](https://arxiv.org/abs/2210.09461). These results show our approach achieves a better balance between speed and accuracy compared to the state-of-the-art.
>
> > **Clarification on Run Time and Latency Measurements**
>
> Thank you for your feedback. To provide a comprehensive evaluation, we have reported a comparison of **throughput** in **Table 1 of our paper**, which shows **the number of images processed per second** by our method and other methods. The testing hardware and setup details are in the "Implementation Details" section: **"we measure the throughput of the models on a single NVIDIA V100 GPU with the batch size fixed to 256."** This provides a direct comparison of performance under the same conditions. We hope this clarifies the effectiveness of our approach not just in terms of theoretical FLOPs but also in practical scenarios on real hardware.
>
> > **Applicability of Our Approach to Swin-like Architectures**
>
> Thank you for your insightful comment. In our paper, we have demonstrated the flexibility and generalization of our approach on different variants of ViTs, including [LV-ViT-S (Jiang et al., 2021)](https://arxiv.org/abs/2104.10858), [T2T-ViT (Yuan et al., 2021)](https://arxiv.org/abs/2101.11986), and [PS-ViT(Yue et al., 2021)](https://arxiv.org/abs/2108.01684), with results shown in Table 2, Figure 5, and Figure 6 (top 2 rows). Among these, LV-ViT-S is a classification model that achieves higher performance than [Swin-S (Liu et al., 2021)](https://arxiv.org/abs/2103.14030) on image classification tasks, highlighting the robustness of our method across different transformer-based architectures.
>
> Regarding the applicability to multi-stage hierarchical architectures of ViT variants, such as [PvT (Wang et al., 2021)](https://arxiv.org/abs/2102.12122) and Swin, our approach can be adapted to these models with slight modifications. These models are characterized by the incorporation of **spatial reduction blocks** within certain layers. ***For these blocks, we retain the attentive tokens from the entire token set at each token compression stage and use masking or padding operations to preserve the full spatial structure during both training and inference***. We have further substantiated this discussion with experimental results under the *off-the-shelf* configuration, as shown in **the last row of Figure 6 of our newly upload paper**.

---

> > ### Comment · Reviewer_ACnx · 2024-10-03
> > **reply**
> >
> > I thank the authors for addressing my comments, especially in the newly added discussions in the updated paper. most of my concerns were addressed

---

### Review · Reviewer_aCeL · 2024-08-27

**Summary Of Contributions:**

The work proposes a heuristic algorithm that combines token pruning and pooling in Vision Transformer for efficient training/inference. The algorithm is able to increase the throughput by up to 50% with minimal degradation.

**Audience:**

Yes

**Broader Impact Concerns:**

Not applicable.

**Claims And Evidence:**

No

**Requested Changes:**

I suggest the authors to change the paper to address the weaknesses, as discussed in the last section.
1. Improve literature survey, including but not limited to the part for vision transformers.
2. Add self-contained presentations of previous methods --- especially for token pruning/pooling.
3. Add experiments for other architectures/scenarios, or at least add a discussion for limitation of the proposed method (why it can't be used in other architectures/scenarios.)

**Strengths And Weaknesses:**

Strengths:
1. The idea of the proposed algorithm is intuitive to understand (despite somewhat poor presentation).
2. The experimental results are solid, which surpass all previous methods reported in the paper.

Weaknesses:
1. The literature survey seems incomplete to me. I am not an expert in token pruning/pooling, but as far as I know the survey for vision transformer missed a few important recent works such as PVT, SWin/CSWin.
2. The presentation of the method assumes that the audience already knows the previous methods well. For example, the definition of attention module is missing. Also, the introduction of token pruning discusses the case when the number of heads equals to one, which does not reflect the reality. For token pooling, the definition of s is not clear (again, it assumes the number of heads is one).
3. The experiment only show case one scenarios (ViT for image classification). I wonder if the method also applies to more advanced architectures (such as PVT, SWin/CSWin) or other computer vision scenarios (segmentation or detection).

---

> ### Author Response · Authors · 2024-09-02
> **Response to Reviewer aCeL**
>
> We sincerely appreciate your feedback. We respond to each of your concerns and suggestions one-by-one in what follows:
>
> > **Improving the Literature Survey and Providing Self-contained Presentations of Previous Methods**
>
> Thank you for your valuable feedback. We have expanded the **"Related Work"** section to provide a more comprehensive overview of the field. Additionally, we have added a new **"Preliminaries"** section to clarify key concepts such as the attention module in vision transformers (ViTs). In the **"Methods"** section, we have also provided more in-depth descriptions of the technical details related to token pruning and pooling.
>
> Regarding your comment on multi-head attention, we did not assume a single head; rather, we applied averaging to perform operations at the token level. We have clarified this point in the revised manuscript, which is now available for review.
>
> > **More experiments for other architectures and scenarios**
>
> We understand the importance of evaluating our method on a broader range of architectures and scenarios. **In the last row of Figure 6 of our newly upload paper, we have included new experiments applying our method to hybrid vision transformers such as [Swin (Liu et al., 2021)](https://arxiv.org/abs/2103.14030) and [PvT (Wang et al., 2021)](https://arxiv.org/abs/2102.12122) to demonstrate the flexibility and generalization of our approach**.
>
> Also, we have extended our experiments to **video classification**, a promising direction for applying our PPT combined with [**Spatiotemporal MAE** (Feichtenhofer et al., 2022)](https://arxiv.org/abs/2205.09113) on the [**Kinetics-400** (Kay et al., 2017)](https://arxiv.org/abs/1705.06950) dataset. We tested both off-the-shelf and with fine-tuning across different FLOPs levels, similar to our approach for image data. These results show our approach achieves a better balance between speed and accuracy compared to the state-of-the-art.

---

> > ### Comment · Reviewer_aCeL · 2024-09-30
> > **Review of the revised version**
> >
> > Thank you for adopting some of my suggestions in the updated draft. However, I think the updates are not enough (more addition than revision). Please see my detailed comments below.
> >
> > 1. Literature survey.
> > While the related works on ViT have been substantially enriched, the paragraph itself becomes too lengthy without a clear outline. Also, the other two paragraphs remain unchanged --- I am not sure whether they have covered more recent works (given that most literatures date back to 2022).
> >
> > 2. Self-contained presentation.
> > The authors added a section on preliminaries. However, the section itself is not self-contained. While the authors talks about MULTI-HEAD self-attention, the whole section assumes the number of heads equals to one.
> >
> > For token pruning/pooling, the problems are not fixed yet. All equations still assume the number of heads is one, and the authors only add explanation in words how to deal with multi-head scenarios. I think making the equations rigorous is not a difficult task given the introduction to MHSA is self-contained.
> >
> > For token pooling, the scalar s is not clearly explained. "s is a row vector that reflects the number of tokens represented and are combined with tokens any time" is hard to understand without math notations.
> >
> > 3. Other architectures: SWin and PVT.
> > These most popular recent architectures are not discussed in the main sections but only briefly mentioned in the experiment section. From the current explanation, I couldn't understand in general how the proposed method deals with spatial dependency (thus not all tokens are treated equal from a specific token's perspective). I suggest the authors add a section to discuss the case (with math rigor).
> >
> > 4. Other scenarios: semantic segmentation and object detection.
> > I asked for these two scenarios in my first review, but the authors added video classification instead. Let me put my question more clearly --- does the proposed method works for dense prediction problems where spatial information for each location is need? I ask this question because the proposed method have to throw away tokens at chosen locations. I wonder if such loss can be remedied for dense prediction problems?
> >
> > In summary, I believe the updated draft is still not ready for publication (many explanations are hand-wavy), I suggest the authors to revise more thoroughly. Please understand that my comments purely focus on the presentation quality, and I don't have specific problems on the idea itself.

---

> ### Author Response · Authors · 2024-10-27
> **Response to Reviewer aCeL**
>
> Thank you for your detailed review and valuable comments. We appreciate your time and effort in evaluating our work.
> Below, we address your concerns point-by-point **in revised manuscript**.
>
> > **Literature Survey**
>
> + We have restructured the related works on **Vision Transformers** to improve readability and clarity by breaking them into thematic subsections.
> + We have updated the discussion to include more recent literature on **token pruning and pooling methods**, particularly works published in 2023 and 2024, ensuring comprehensive and up-to-date coverage.
>
> > **Self-Contained Presentation**
>
> + We have updated the description of **multi-head self-attention (MHSA)** to explicitly account for multiple heads. Specifically, we revised the equations and clarified how the queries, keys, and values are computed **across different heads**. This ensures the presentation is rigorous and does not assume a single-head scenario.
> + For token pruning, we also updated the corresponding equations for the **multi-head scenario**.
> + For token pooling, we further **clarified the role of the scale $s$ with mathematical notations** and explained how it adjusts the attention scores based on the size of each token.
>
> > **Other Architectures**
>
> We have added a new section, **"Adaptation to Multi-Stage Hierarchical Architectures"**, discussing how our approach can be adapted to architectures like Swin and PvT using relevant mathematical equations.
>
> > **Other Scenarios**
>
> Recent studies ([DToP, ICCV'23](https://arxiv.org/abs/2308.01045), [Focus-DETR, ICCV'23](https://arxiv.org/abs/2307.12612), [AiluRus, NeurIPS'24](https://arxiv.org/abs/2311.01197)) have shown that **it is feasible to handle dense prediction tasks like segmentation and detection while using token sparsification techniques**. However, our paper primarily compares against classical token sparsification methods (e.g., DynamicViT, E-ViT, ToMe) on ImageNet. We have additionally verified our method across a range of models and tasks, demonstrating its generality. We believe that adapting our method to dense prediction tasks, such as detection and segmentation, presents a promising direction for future research, which we plan to explore in subsequent work.

---

### Review · Reviewer_gChe · 2024-09-22

**Summary Of Contributions:**

This paper introduces the Pruning & Pooling Transformer (PPT) framework, a method to enhance the efficiency of Vision Transformers (ViTs) by addressing two types of redundancy: inattentive and duplicative. PPT adaptively applies token pruning to discard less relevant tokens and token pooling to merge similar tokens, optimizing performance across different layers without adding extra trainable parameters. Extensive experiments show that PPT outperforms state-of-the-art token compression techniques, offering a superior accuracy-efficiency trade-off.

**Audience:**

Yes

**Claims And Evidence:**

No

**Requested Changes:**

Adding results from some intuitive method combinations of token pruning and pooling. I also encourage the authors to strengthen their motivation and contribution.

**Strengths And Weaknesses:**

Strengths:

1: The method doesn't introduce extra trainable parameters, allowing seamless integration with pre-trained ViTs.

2: PPT performs well across various Vision Transformer models and tasks, showing its versatility and robustness.

3: PPT intelligently applies token pruning in deeper layers and token pooling in shallow layers based on token importance and redundancy, optimizing efficiency dynamically.


Weaknesses:

1: The whole method seems simple, combining token pooling and pruning. In other words, practically, we can perform token pooling first and prune the pooled tokens later. How about the results of this baseline? I haven't found solid motivation for the importance of the proposed method over the simple combination.

2: The results are not solid and look trivial. As shown in Table 1, the improvement is very limited (0.1 top-1 ACC on ImageNet) compared to the previous SOTA TPS. Similar situations can also be found in EViT v.s. PPT on DeiT-B, Table 2 and Figure 3. It's hard for me to convince the effectiveness of the proposed method.

3: The backbone used for comparison is too weak (DeiT). How about the results on the Swin Transformer or other more advanced transformer backbone?

*I reviewed the latest version (17 Sep).*

---

> ### Author Response · Authors · 2024-09-23
> **Response to Reviewer gChe**
>
> We sincerely appreciate your feedback. We respond to each of your concerns and suggestions one-by-one in what follows:
>
> > **Comparative Analysis of Token Pooling and Pruning Integration**
>
> Thank you for your valuable feedback. We have addressed this in our ablation study **"Effectiveness of Techniques Integration" (Figure 8)**, where we compare **our method (red line)** with the **suggested approach (yellow line)**. While combining the two techniques within a block demonstrates benefits at lower FLOPs, it compromises token pooling performance at higher FLOPs. In contrast, our framework consistently achieves optimal performance across varying FLOPs, showcasing its overall effectiveness and superiority in balancing performance and computational efficiency.
>
> > **The Effectiveness of the Proposed Method**
>
> Thank you for your insightful comment. As shown in Table 1, we followed settings used in many prior works, reporting accuracy based on specific FLOPs. In this high-FLOPs range, many methods closely approach the original model's accuracy. While our method achieves state-of-the-art performance, the improvement may appear marginal.
>
> To provide a broader perspective, Figure 4 compares various FLOPs using DeiT-S as the baseline. Our method demonstrates superiority under different conditions:
>
> - **Lower FLOPs (<2.5G):** Our method exhibits a noteworthy performance improvement of 0.7%-3.8% compared to other pruning methods.
> - **Higher FLOPs (>3.0G):** Our method demonstrates superior performance compared to the baseline model, surpassing the capabilities of the comparison methods.
> - **Off-the-shelf (without fine-tuning):** PPT demonstrates significant improvement (0.1%-2.3%) compared to other methods. Notably, PPT also achieves competitive results with other fine-tuned models.
>
> These results underscore the robustness and versatility of our approach.
>
> > **Applicability of Our Approach to More Advanced Architectures**
>
> Thank you for your insightful comment. In our paper, we demonstrated the flexibility and generalization of our approach across different ViT variants, including [LV-ViT-S (Jiang et al., 2021)](https://arxiv.org/abs/2104.10858), [T2T-ViT (Yuan et al., 2021)](https://arxiv.org/abs/2101.11986), and [PS-ViT (Yue et al., 2021)](https://arxiv.org/abs/2108.01684), as shown in Table 2, Figure 5, and Figure 6. LV-ViT-S, in particular, achieves higher performance than [Swin-S (Liu et al., 2021)](https://arxiv.org/abs/2103.14030) in image classification tasks, highlighting the robustness of our method across transformer architectures.
>
> Regarding its adaptability to multi-stage hierarchical architectures, such as [PvT (Wang et al., 2021)](https://arxiv.org/abs/2102.12122) and [Swin (Liu et al., 2021)](https://arxiv.org/abs/2103.14030), our approach can be integrated with slight modifications. These architectures utilize **spatial reduction blocks**, where ***we retain attentive tokens from the entire token set during each compression stage, using masking or padding to maintain spatial structure during training and inference***.
>
> This discussion is further supported by our experiments in the *off-the-shelf* configuration, shown in **the last row of Figure 6 in the newly uploaded paper (uploaded on September 2nd)**.

---

> > ### Comment · Reviewer_gChe · 2024-09-24
> >
> > Thanks for the response from the authors. I appreciate the efforts.
> >
> > I have a question. For the first weakness I raised, if I understand correctly, is this paper's main contribution the "Adaptive Token Pruning & Pooling Strategy" shown on Page 6?

---

> > > ### Author Response · Authors · 2024-09-24
> > > **Response to Reviewer gChe**
> > >
> > > Thank you for your thoughtful question. While the "Adaptive Token Pruning & Pooling Strategy" on Page 6 is indeed a key part of our contribution, the main novelty lies in integrating both techniques into a unified framework through a heuristic approach.
> > >
> > > In addition, our contributions include:
> > >
> > > + We provide an analysis of the variance in importance scores across different layers and samples within ViTs, revealing why our method is effective and explaining the rationale behind our design.
> > >
> > > + Extensive experiments show promising results across various ViT variants and datasets, demonstrating the robustness and versatility of our approach.
> > >
> > > We hope our PPT could bring a new perspective for obtaining efficient vision transformers.

---

### Review · Reviewer_x5Aj · 2024-10-07

**Summary Of Contributions:**

This paper proposes a method to improve the efficiency of vision transformers (ViTs), which is known as Pruning & Pooling Transformer. It works by combining token pruning and pooling -- the key idea is to compute a "significance score" for each token and to perform token pruning if the variance of the significance scores is larger than a threshold, and token pooling otherwise. The significance score is based on the attention weights assuming that the query is the classification token, and the norms of the values of each token. Experiments show that the proposed method achieves comparable results to the baselines in terms of accuracy and throughput.

**Audience:**

Yes

**Broader Impact Concerns:**

No concerns

**Claims And Evidence:**

No

**Requested Changes:**

I would suggest adding a motivation for the particular choice of significance score and a justification of why it makes sense for the significance score to not account for the variance of the queries and the scale of the weights in the linear layer. Additionally, I would suggest adding the results of the proposed method at different points of the accuracy vs. throughput tradeoff to Table 1, so that definitive conclusions can be drawn on the effectiveness of the proposed method compared to the baselines. Moreover, I would suggest adding baselines that don't require fine-tuning and baselines that are not ViT-specific, and discussing the broader literature on efficient transformers that aren't ViT-specific as part of the related work.

**Strengths And Weaknesses:**

Strengths:

- The idea is simple and reasonably well-motivated.
- The method can be used without training/fine-tuning the base ViT.

Weaknesses:

- The significance score for the tokens may not be sensible and could mislead. For example, the significance score does not account for the variance of the queries -- when it's high, the attention weights using the classification token as the query can be very different from the attention weights using other tokens as queries. As a result, the significance score may not reflect what the actual attention weights are. The significance score is also sensitive to parameterization -- for example, it depends on the scale of the values of each token, but not the scale of the weights in the linear layer. As a result, if we were to consider two different ViTs, one of which has a larger scale of the values but smaller scale of the weights, and the other of which has a smaller scale of the values but larger scale of the weights, token pruning is more likely to be picked for the former than the latter even though the former may compute the same thing as the latter. I am therefore concerned about the technical soundness of the proposed method.
- As shown in Table 1, compared to other baselines, the proposed method seems to be better in accuracy, but worse in throughput for both DeiT-Ti and DeiT-B. This could mean that the method is making a different tradeoff in accuracy vs. throughput compared to the baselines, and so it doesn't show whether the proposed method is better or worse than the baselines. Hence, the results are inconclusive, and it's unclear whether the proposed method performs better or worse than the baselines.
- Comparisons were only made to baselines that require fine-tuning, even though one of the claimed contributions of the proposed method is not requiring fine-tuning.
- Comparisons were only made to ViT-specific methods, rather than general efficient transformer methods that apply to other kinds of transformers as well.
- Similarly, the literature review only discusses ViT-specific methods and makes no mention of general-purpose methods that apply to other kinds of transformers.

---

> ### Author Response · Authors · 2024-10-24
> **Response to Reviewer x5Aj (I)**
>
> We sincerely appreciate your feedback. We respond to each of your concerns and suggestions one-by-one in what follows:
> > **Motivation for the particular choice of significance score**
>
> Thank you for your thorough review and valuable comments on our work. You highlighted that our significance score calculation might not consider *the variance of the query vectors* and that the score may be *sensitive to parameterization*. These are indeed important considerations.
> + **Rationale Behind Our Design**
>
> Since only the classification token directly impacts the final prediction, our focus is primarily on it. The output of the classification token
>   $O_{cls}$  is computed as:  $O_{cls} = \sum_{i=1}^{N+1} A_{1,i} \times V_{i} $, where $A_{1,i}$ represents the attention weight between the classification token and token $i$, and $V_{i}$ is the value vector of token $i$. This equation demonstrates that $O_{cls}$ is a weighted sum of the values of all $N+1$ tokens (the first token is the classification token itself), with the weights determined by attention scores that indicate the relevance of each token to the classification token.
>
> Intuitively, the norm of the weighted value vector reflects each token's contribution and importance to the final prediction. Therefore, we define the significance score of image token $i$ as: $Score_i = \frac{ A_{1,i+1} \times  ||V_{i+1}||} {\sum_{j=1}^{N} A_{1,j+1} \times ||{V}_{j+1}||} \quad i, j \in \{1, \ldots, N\}$, excluding the classification token itself from the pruning process. This formulation captures both the attention weight and the value magnitude, aligning with the token's influence on the classification outcome.
> The effectiveness of this design has been validated in previous token pruning methods such as [DynamicViT](https://arxiv.org/abs/2106.02034), [EViT](https://arxiv.org/abs/2202.07800), and [ATS](https://arxiv.org/abs/2111.15667), which also utilize attention-based significance measures for token selection.
>
> + **Response to Concern: Not consider the Variance of Query Vectors**
>
> Thank you for pointing out the potential impact of query vector variance. To address this, we redesigned the significance score to better reflect the influence of query variance. Instead of only considering the attention scores from the classification token, we now aggregate the attention scores over all query vectors to evaluate each token's importance. The new significance score for image token $i$ is given by: $Score_{i} = \sum_{j=1}^{N} A_{i+1, j}, \quad i, j \in \{1, \ldots, N\}$, where $\mathcal{A}_{i, j}$ represents the attention score between query vector $i$ and key vector $j$. By summing over all query vectors, this new metric comprehensively captures each token's contribution from all queries.
>
> We conducted experiments comparing the model performance using the newly designed score versus the original one. The results are summarized below:
>
> |FLOPs (G)|New Metric|Original Metric (Ours)|
> |:-:|:-:|:-:|
> |2.7|79.05|**79.13**|
> |2.9|79.39|**79.49**|
> |3.5|79.68|**79.71**|
>
> The results indicate that the model's performance using the new score, which fully accounts for query variance, is comparable to the original. The differences in accuracy are minimal, with the original metric slightly outperforming the new one. This suggests that while our original score captures token importance effectively, the new score provides a more holistic view without significantly altering performance. These findings demonstrate the robustness of our original approach.
> + **Response to Concern: Score May be Sensitivity to Parameterization:**
>
> Thank you for pointing out the potential sensitivity of the significance score to parameterization. We understand the concern that the score could depend on the scale of value vectors, making it sensitive to parameter settings.
>
> In our ablation study (see "Different token pruning policy" and Figure 10), we explored whether incorporating the value vector norm affects the significance score. The results indicate that including the value vector has a negligible effect in the off-the-shelf setting but can slightly improve performance with fine-tuning. This suggests that our score is relatively robust, with the value vector contributing marginally to performance enhancement.
>
> By applying **Layer Normalization** before computing query, key, and value vectors, we ensure consistent input scales, reducing sensitivity to parameter variations. Our results show that token pruning decisions remain consistent and model performance is robust across different parameter settings, further validating the soundness of our method.

---

> ### Author Response · Authors · 2024-10-24
> **Response to Reviewer x5Aj (II)**
>
> > **Results of the Proposed Method under different FLOPs**
>
> Thank you for your insightful comments on the tradeoff between accuracy and throughput, as highlighted in Table 1. We recognize that while our method achieves better accuracy compared to the baselines, it has lower throughput, reflecting a different tradeoff.
>
> In Table 1, we followed the settings used in prior works, reporting accuracy based on specific FLOPs, and our method achieves state-of-the-art performance in this context. While some methods achieve higher throughput, but this often comes at a significant loss in accuracy. Our method, on the other hand, offers a more balanced tradeoff, prioritizing accuracy without sacrificing too much efficiency.
>
> In Figure 4, we provides a broader comparison of FLOPs using DeiT-S as the baseline. Our method demonstrates consistent superiority:
>
> - **Lower FLOPs (<2.5G):** 0.7%-3.8% improvement over other techniques.
> - **Higher FLOPs (>3.0G):** Superior performance compared to the baseline.
> - **Off-the-shelf (without fine-tuning):** 0.1%-2.3% improvement, competitive with fine-tuned models.
>
> These results underscore the robustness and versatility of our approach.
> > **Comparisons to Off-the-shelf Baselines**
>
> Thank you for your comment regarding the comparison to baselines that do not require fine-tuning. We acknowledge that one of the key contributions of our method is its effectiveness in the off-the-shelf setting (*i.e.,* without fine-tuning).
>
> In Figure 4 (right) and in the supplementary material, we provided a detailed comparison of our method with other token sparsification techniques under the off-the-shelf setting. The results clearly demonstrate the superiority of our approach compared to other methods that do not require fine-tuning. Specifically, our method achieves 0.1%-2.3% improvement over other baselines, even without the need for fine-tuning, showcasing its robustness and versatility in scenarios where fine-tuning is not feasible.
> > **Applicability of Our Approach to More Advanced Architectures**
>
> Thank you for your insightful comment. In our paper, we demonstrated the flexibility and generalization of our approach across different ViT variants, including [LV-ViT-S (Jiang et al., 2021)](https://arxiv.org/abs/2104.10858), [T2T-ViT (Yuan et al., 2021)](https://arxiv.org/abs/2101.11986), and [PS-ViT (Yue et al., 2021)](https://arxiv.org/abs/2108.01684), as shown in Table 2, Figure 5, and Figure 6. LV-ViT-S, in particular, achieves higher performance than [Swin-S (Liu et al., 2021)](https://arxiv.org/abs/2103.14030) in image classification tasks, highlighting the robustness of our method across transformer architectures.
>
> Regarding its adaptability to multi-stage hierarchical architectures, such as [PvT (Wang et al., 2021)](https://arxiv.org/abs/2102.12122) and [Swin (Liu et al., 2021)](https://arxiv.org/abs/2103.14030), our approach can be integrated with slight modifications. These architectures utilize **spatial reduction blocks**, where ***we retain attentive tokens from the entire token set during each compression stage, using masking or padding to maintain spatial structure during training and inference***.
>
> This discussion is further supported by our experiments in the *off-the-shelf* configuration, shown in **the last row of Figure 6** in the newly uploaded paper.

---

### Decision · Action_Editor_a2GE · 2024-10-28

**Recommendation:** Reject

**Comment:**

The main goal of this paper is to reduce the computational complexity of the model while maintaining accuracy. However, as pointed out by the reviewers, the authors have not sufficiently provided corresponding arguments in the experiments. While accuracy improves, throughput is worse than the baselines, making its impact uncertain. The proposed significance score may not effectively determine token necessity. Additionally, the experiments focus only on one scenario (ViT for image classification).
The authors’ claim that pruning and pooling at the token level can maintain accuracy lacks experimental or theoretical support. In conclusion, this paper lacks evidence for its motivation and method effectiveness.

**Audience:**

The topic of this article is aimed at researchers engaged in pruning. The related conclusions are uncertain, making it less likely to capture the readers' interest.

**Claims And Evidence:**

1.  This paper argues that it is not optimal to either only reduce inattentive redundancy by token pruning or only reduce duplicative redundancy by token merging. This argument is not fully investigated or strictly proved.
2. The authors claimed that the proposed PPT reduces complexity and improves throughput without any accuracy drop on the ImageNet dataset. The authors provided corresponding empirical results. However, the experiments do not perfectly align with this statement. While accuracy improves, throughput is worse than the baselines, making its impact uncertain.